# CharFxReg : Characteristic Function based Regularisation

## ABSTRACT

Regularization plays a crucial role in neural network training by preventing over-fitting and improving generalization. In this paper, we introduce a novel regularization technique grounded in the properties of characteristic functions, leveraging assumptions from decomposable distributions and the central limit theorem. Rather than replacing traditional regularization methods such as L2 or dropout, our approach is designed to supplement them, providing a contextual delta of generalization. We demonstrate that integrating this method into standard architectures improves performance on benchmark datasets by preserving essential distributional properties and mitigating the risk of overfitting. This characteristic function-based regularization offers a new perspective in the direction of distribution-aware learning in machine learning models.

## 1 INTRODUCTION

Regularization remains a cornerstone of modern machine learning, enabling models, especially deep neural networks with vast parameter spaces, to generalize effectively and avoid overfitting. Traditional norm-based penalties, such as L2 (ridge) (Hoerl & Kennard, 1970), L1 (lasso) (Tibshirani, 1996), and Elastic Net, have long been foundational for controlling model complexity and promoting sparsity. Meanwhile, stochastic regularization techniques like dropout (Srivastava et al., 2014) and its variants mitigate co-adaptation among units, and implicit methods including early stopping (Prechelt, 1998), data augmentation, and label perturbation have further enriched this landscape. Structured (Vapnik, 2006) and group-wise regularizers exploit dependencies among parameters to yield more interpretable and computationally efficient models. Collectively, these methods form a rich, well-studied toolkit that underpins much of contemporary machine learning.

In parallel, a growing body of work in probability and statistics has explored characteristic function–based methods, leveraging the Fourier transform of probability distributions to overcome challenges posed by intractable or unknown density functions (Warr, 2014). These techniques have established rigorous computability results (Mori et al., 2009), advanced numerical methods for quantile estimation (Junike, 2025), inversion methods for density estimation (Lunde et al., 2018), enabled efficient simulation of stochastic processes (Boyarchenko & Levendorskii, 2023), and led to practical software tools supporting broad applicability (Witkovsky, 2024). Applications of characteristic function–based approaches span jump-diffusions, semi-Markov models, Lévy processes, and more.

Despite these advances, characteristic function–based methods have received limited attention within the machine learning community, particularly in the context of regularization and model generalization. While classical regularization methods primarily operate in the parameter or output space, characteristic function–based inference offers a complementary perspective rooted in distributional structure and transform-domain analysis. This suggests potentially intriguing parallels that remain largely unexplored.

This paper seeks to explore this gap by investigating regularization through the lens of characteristic function–based methods. We explore how characteristic functions can serve as regularizers that implicitly incorporate distributional information, providing new ways to control model behavior and improve generalization. Our approach offers a principled yet simple framework that complements existing regularization techniques by exploiting the rich representational power of characteristic functions.

Our main contribution is the establishment of a novel regularization framework that leverages characteristic function–based methods to impose distributionally informed constraints on model training. By integrating characteristic function perspectives with classical regularization principles, we open new avenues for incorporating probabilistic structure into machine learning models. This work lies at the intersection of transform-based inference and regularization, with the goal to enhance model robustness and performance.

## 2 RELATED WORK

Traditional approaches such as L1 and L2 regularization (Tibshirani, 1996; Hoerl & Kennard, 1970) constrain model complexity by penalizing the magnitude or sparsity of weights. In deep learning, stochastic methods like dropout (Srivastava et al., 2014) and data-augmentation-based techniques like mixup (Zhang et al., 2017) inject noise or synthetic variability to guide learning toward more robust solutions. Other strategies operate implicitly, such as early stopping (Prechelt, 2002) and sharpness-aware minimization (Foret et al., 2020), which regularize via optimization dynamics. However, these methods primarily affect the learning process in parameter space, offering limited interpretability in terms of output distributional behaviour.

An emerging line of work studies the statistical properties of a model's *output distribution* as a lens for regularization and evaluation. Generative models frequently use divergences between empirical and target distributions e.g., Kullback-Leibler, Jensen-Shannon, or Wasserstein metrics, as objectives or diagnostics (Goodfellow et al., 2014; Arjovsky et al., 2017). Other approaches employ *kernel-based distances* like Maximum Mean Discrepancy (MMD) (Gretton et al., 2012) or energy distances (Székely et al., 2004), which implicitly rely on moment-matching or characteristic function alignment. Despite these advances, the explicit use of *central limit theorem (CLT)* assumptions to model and regularize output distributions is rare. While CLT-based arguments have been invoked to explain model behavior in wide neural networks (Lee et al., 2017; Matthews et al., 2018), they are seldom used as direct regularization tools.

Our methodology rests on the premise that the final layer outputs of the network can be modeled as a collection of Bernoulli random variables, whose aggregate behaviour, by virtue of the Lyapunov Central Limit Theorem (Lyapunov, 1900), approaches a Gaussian distribution in the limit of large dimensionality. Rather than enforcing normality on parameters or intermediate activations, we focus on the distribution of these output sums, leveraging their probabilistic structure to define a natural frequency-domain regularizer. Deviations from the predicted Gaussian characteristic function serve as an informative metric quantifying the network's departure from the asymptotic regime, thus reflecting potential overfitting or instability in representation learning. This framework imposes a functional prior that guides gradient updates toward outputs exhibiting asymptotic normality, effectively harnessing classical limit theorems within the modern paradigm of deep learning. To our knowledge, this represents a novel instantiation of Lyapunov CLT-based constraints as a regularizing force acting directly on neural network training.

## 3 REGULARIZATION METHODOLOGY

We now propose a regularization framework that leverages distributional convergence to enforce structured behavior in neural network outputs.

### 3.1 MOTIVATION AND HIGH-LEVEL INTUITION

Specifically, we aim to softly constrain the model's output representations by encouraging them to relax toward a target distribution, achieved by modeling the output layer as a function of Bernoulli random variables. This modeling is motivated by the observation that, in many classification settings, the output of a neural network (especially under sigmoid or softmax activation) can be interpreted as a sequence of Bernoulli trials. Each output unit reflects a probabilistic decision, and collectively, these outputs can be visualized as a chain of Bernoulli outcomes. By slightly reformulating this output structure, we construct a random variable as described in Definition 3, which allows us to treat the final layer's activations as a sample from an approximate sum of independent Bernoulli variables.

Under this formulation, and by invoking the Lyapunov Central Limit Theorem, we establish that this random variable converges in distribution to the standard normal distribution as the number of Bernoulli components increases and certain conditions are met. This theoretical insight forms the basis for a new regularization term that penalizes deviations from this target Gaussian distribution. Practically, this is implemented by comparing the characteristic function of the empirical representation with that of the standard normal, and incorporating the resulting distance into the loss function. Because the construction of $\aleph$ is flexible, the regularizer can be generalized to different layers or structural parts of the network. In this work, however, we focus on applying the regularization at the output layer to keep the formulation both interpretable and practically feasible, while demonstrating strong generalization improvements.

We now proceed to derive the regularization. The starting point is the Lyapunov Central Limit Theorem, which establishes that suitably normalized sums of independent random variables converge in distribution to a Gaussian under mild moment conditions.

## 3.2 Formalising the Regularization Approach

**Definition 1** (Lyapunov Central Limit Theorem). Suppose we have a sequence of independent random variables, $\{Y_1, Y_2, \ldots, Y_n\}$, each with finite expected value $\mu_i$ and variance $\sigma_i^2$.

If we define the following sum of variances:

$$s_n^2 = \sum_{i=1}^{n} \sigma_i^2. \tag{1}$$

If $\exists \delta > 0$, such that Lyapunov's condition:

$$\lim_{n \to \infty} \frac{1}{s_n^{2+\delta}} \sum_{i=1}^{n} \mathbb{E}\left[|Y_i - \mu_i|^{2+\delta}\right] = 0, \tag{2}$$

is satisfied $\implies$ the sum of the normalized variables $\frac{Y_i - \mu_i}{s_n}$ converges in distribution to a standard normal random variable as $n \to \infty$:

$$\frac{1}{s_n} \sum_{i=1}^{n} (Y_i - \mu_i) \to N. \tag{3}$$

where $N \sim \mathcal{N}(0, 1)$
(Lyapunov, 1900)

We model each data point as being generated from a random variable $\aleph$, which is represented as an approximation of linear combination of Bernoulli variables $X_i \sim \text{Bern}(p)$. We will now demonstrate that properly formulating $\aleph$ allows for convergence to $\mathcal{N}(0, 1)$ as the number of Bernoulli variables increases sufficiently.

**Axiom 1.** Let us establish the following fundamental assumption that will underpin our framework. Given the true data distribution, $\mathcal{D}$ (which can be conceptualized as the "Population Distribution" in statistical terms), we assert that the data points (the "Samples" we usually have in our finite dataset) are generated from a random variable $\aleph$ such that:

$$\aleph \sim \mathcal{D} \tag{4}$$

**Definition 2.** The characteristic function $\phi(u)$ of a random variable $Y$ is defined as:

$$\phi(u) = \mathbb{E}[e^{\vartheta u Y}].^1 \tag{5}$$

**Definition 3.** We model the random variable $\aleph$, as a linear combination of Bernoulli Random Variables, defined as follows:

$$\aleph = \frac{1}{s_n} \sum_{i=1}^{n} (X_i - \mu_i), \tag{6}$$

where $X_i \sim Bern(p_i) \implies \mu_i = \mathbb{E}[X_i] = p_i$ and $s_n^2 = \sum_{i=1}^{n} Var[X_i] = \sum_{i=1}^{n} p_i(1 - p_i).$

---

[1] We deviate from common practice and use $\vartheta$ instead of $i$ to define the imaginary unit in a bid to reduce confusion as the letter $i$ is used for indexing in much of the later proofs and writing

### 3.3 CHARACTERIZING THE OUTPUT DISTRIBUTION

**Proposition 1.** The characteristic function for $\aleph$ can be computed as follows from 5 and 6:

$$\phi_{\mathcal{D}}(u) = \prod_{i=1}^{n} e^{\frac{(-\vartheta u p_i)}{\sqrt{\sum_{i=1}^{n}(p_i*(1-p_i))^2}}} (1-p_i) + \prod_{i=1}^{n} e^{\frac{(\vartheta u(1-p_i))}{\sqrt{\sum_{i=1}^{n}(p_i*(1-p_i))^2}}} (p_i) \tag{7}$$

*Proof.*

$$\phi_{\mathcal{D}}(u) = \mathbb{E}[e^{\vartheta u \aleph}] = \mathbb{E}\left[e^{\vartheta u \frac{1}{s_n}\sum_{i=1}^{n}(X_i-\mu_i)}\right]. \tag{8}$$

Using the product law of exponents ($a^n * a^m = a^{(n+m)}$), we rewrite the characteristic function:

$$= \mathbb{E}\left[\prod_{i=1}^{n} e^{\vartheta u \frac{1}{s_n}(X_i-\mu_i)}\right]. \tag{9}$$

Next, we separate it into the following by linearity of the expectation:

$$= \mathbb{E}\left[\prod_{i=1}^{n} e^{\vartheta u \frac{1}{s_n}(X_i-\mu_i)}\mathbb{I}\{X_i=0\}\right] + \mathbb{E}\left[\prod_{i=1}^{n} e^{\vartheta u \frac{1}{s_n}(X_i-\mu_i)}\mathbb{I}\{X_i=1\}\right]. \tag{10}$$

By properties of the Bernoulli Random Variable this is more precisely:

$$= \mathbb{E}\left[\prod_{i=1}^{n} e^{\vartheta u \frac{1}{s_n}(0-p_i)}\mathbb{I}\{X_i=0\}\right] + \mathbb{E}\left[\prod_{i=1}^{n} e^{\vartheta u \frac{1}{s_n}(1-p_i)}\mathbb{I}\{X_i=1\}\right]. \tag{11}$$

Using linearity of expectation, we have:

$$= \prod_{i=1}^{n} e^{\vartheta u \frac{1}{s_n}(-p_i)}\mathbb{E}[\mathbb{I}\{X_i=0\}] + \prod_{i=1}^{n} e^{\vartheta u \frac{1}{s_n}(1-p_i)}\mathbb{E}[\mathbb{I}\{X_i=1\}]. \tag{12}$$

By definition of Expectation of Indicator Function, we have:

$$= \prod_{i=1}^{n} e^{\vartheta u \frac{1}{s_n}(-p_i)}\mathbb{P}(X_i=0) + \prod_{i=1}^{n} e^{\vartheta u \frac{1}{s_n}(1-p_i)}\mathbb{P}(X_i=1). \tag{13}$$

By properties of the Bernoulli Random Variable, this can be re-expressed as:

$$= \prod_{i=1}^{n} e^{\vartheta u \frac{1}{s_n}(0-p_i)}(1-p_i) + \prod_{i=1}^{n} e^{\vartheta u \frac{1}{s_n}(1-p_i)}p_i. \tag{14}$$

Reformulating the $s_n^2$:

$$\because s_n^2 = \sum_{1}^{n}\sigma_i^2 \Longrightarrow \therefore s_n = \sqrt{\sum_{1}^{n}\sigma_i^2} = \sqrt{\sum_{1}^{n}(p_i(1-p_i))^2} \tag{15}$$

Thus, we conclude:

$$\Longrightarrow \therefore \phi_{\mathcal{D}}(u) = \prod_{i=1}^{n} e^{\frac{(-\vartheta u p_i)}{\sqrt{\sum_{i=1}^{n}(p_i(1-p_i))^2}}} (1-p_i) + \prod_{i=1}^{n} e^{\frac{(\vartheta u(1-p_i))}{\sqrt{\sum_{i=1}^{n}(p_i(1-p_i))^2}}} (p_i) \tag{16}$$

$$\square$$

**Corollary 1.** By the Lyapunov Central Limit Theorem, as $n \to \infty$, the characteristic function converges to that of the characteristic function of the normal distribution:

$$\because \mathcal{D} \to \mathcal{N}(0,1) \implies \therefore \phi_{\mathcal{D}}(u) \to \phi_{\mathcal{N}(0,1)}(u). \tag{17}$$

Note this is true because rate of growth of the moments is contrained as per the Lyapunov condition, described in detail by proof outlined in the appendix A.1. Some graphics from numerical simulation of this effect is also attached in the appendix 7, along with some helper graphs to visualise some transformations of characteristic functions as it is difficult to find some and there aren't really many online.

### 3.4 Formulating the Regularization Objective

**Definition 4** (Regularization). Regularization is a technique used to prevent overfitting by adding a penalty term to the loss function. The regularized loss function is typically expressed as:

$$\min \sum_{i=1}^{n} L(\hat{y}_i, y_i) + \lambda R(f) \tag{18}$$

where $\hat{y}_i = f(x_i)$ is the predicted output, $L$ is the loss function, $R(f)$ is the regularization term, and $\lambda$ is a hyperparameter that controls the trade-off between model fit and complexity.

If we interpret $\phi_{\mathcal{N}(0,1)}$ as a relaxed fit that our function can be adjusted towards, we can establish a regularization term $R(f)$, which levies a penalty on the complexity of model $f$, by adding a constraint through examining the difference between $\phi_D$ and $\phi_{\mathcal{N}(0,1)}$. This can be achieved by measuring the distance between the signals using:

$$\mathcal{R}(f) = d(\phi_D, \phi_{\mathcal{N}(0,1)}) \tag{19}$$

The choice of the distance metric , $d(\cdot)$, is up to the practitioner but we briefly mention the ones we used for evaluation in the appendix B for reference.

### 3.5 Extension to Broader Network Structures

For greater generality, the regularization approach can be extended to different components of the network or to various classes of networks by appropriately redefining the construction of the random variable introduced in Definition 3. This, in turn, accordingly allows for a reformulation of the equation in Proposition 1. In this work, we focus specifically on the output layer formulation as presented and will evaluate its effectiveness on real-world datasets in the following section.

## 4 Experiments

For evaluating the effectiveness of our regularization framework, we begin by considering five main cases: no regularization (None), classical $L^1$, $L^2$, and our proposed $\psi_1$ and $\psi_2$ regularization terms (that are described in detail in Appendix B), which are designed to reflect underlying signal structure. For clarity of exposition, and to focus on the most representative behaviors, we simplify this comparison to **three main regimes**. The first is **ElasticNet** (Zou & Hastie, 2005), which interpolates between $L^1$ and $L^2$ regularization and serves as a standard mixed-norm baseline. The second is our mixed-norm approach called **SpectralNet** ($\psi_{\text{spec}}$) (see Appendix B), which adopts the same formal structure as ElasticNet, but replaces the classical norms with $\psi$-based distances. The third regime corresponds to the **unregularized** case, included as the baseline for context.

For our experiments, we evaluate the proposed regularization method across a diverse set of multi-class classification datasets, spanning various domains including tabular data, text, audio, and images. Table 1 summarizes the key characteristics of these datasets along with the representative models employed, where the output layer size corresponds to the number of classes. The datasets vary widely in scale and complexity, ranging from small tabular datasets like PhiUSIIL and Wine, to large-scale image recognition benchmarks such as ImageNet-1K and ImageNet-21K, as well as audio and text datasets with thousands of classes.

This broad selection allows us to comprehensively assess the generalization capability and robustness of our regularization approach across different modalities and model architectures, including logistic regression, multi-layer perceptrons (MLPs), convolutional neural networks (CNNs), transformers, and state-of-the-art models like BERT and Vision Transformers. Detailed results and analyses on these datasets are provided in the following sections, evaluating the effectiveness of the proposed regularization method in diverse learning scenarios and data modalities in real-world, multi-class classification settings.

Additionally, for all models, core hyperparameters such as regularization parameter $\lambda$ , learning rate were optimized using Optuna (Akiba et al., 2019). Model-specific parameters, such as the

Table 1: Multi-Class Classification Datasets and Representative Models used for Experiments

| # Classes | Dataset Name | Domain | Accuracy Metric | Model (Output Layer = # of Classes) | Notes |
|---|---|---|---|---|---|
| 2 | PhiUSIIL (Phishing URL) | Text | Top-1 | MLP | Binary classification of phishing vs. benign URLs |
| 3 | Waveform | Audio | Top-1 | Logistic Regression | Synthetic waveforms in three categories |
| 5 | BBC News | Text | Top-1 | Multinomial Logistic Regression | News articles in five categories |
| 10 | MNIST | Image | Top-1 | LeNet5 | Handwritten digits (0–9) |
| 15 | Human Action Recognition (HAR) | Image | Top-1 | CNN | Image/video-based human activity recognition |
| 20 | 20 Newsgroups | Text | Top-1 | BERT | Text classification of 20 news categories |
| 35 | Google Speech Commands (GSC) v2 | Audio | Top-1 | RNN | Voice command recognition from audio clips |
| 43 | German Traffic Sign Recognition Benchmark (GTSRB) | Image | Top-1 | VGG19 | Classification of traffic sign images |
| 50 | ESC-50 (Environmental Sound Classification) | Audio | Top-1 | CNN | Environmental audio sounds (e.g., dog bark, siren) |
| 100 | CIFAR-100 | Image | Top-1 | ResNet18 | Tiny natural images across 100 fine-grained classes |
| 196 | Stanford Cars Dataset | Image | Top-1 | EfficientNetB7 | Car classification |
| 251 | FoodX-251 | Image | Top-1 | MobileNetV3 | Food image classification with many categories |
| 500 | Oxford-BBC Lip Reading in the Wild (LRW) Dataset | Image | Top-1 | ResNet-18 + Bi-GRU | Lipreading Dataset |
| 1,000 | CAS-VSR-W1k | Image | Top-1 | ResNet-18 + Bi-GRU | Lipreading Dataset extended from LRW Dataset |
| 5,000 | WebVision 2.0 | Image | Top-5 | ResNet152 | Large-scale noisy web data |
| 10,000 | iNaturalist2021 | Image | Top-5 | EfficientNetB1 | Species classification from real-world observations |

number of hidden layers, number of hidden units per layer, convolutional kernel size and stride (for CNNs), number of attention heads and transformer layers (for Transformers), activation functions, and optimizer type, were also tuned where applicable, ensuring that the reported results, presented in Table 2, reflect the best performance achievable under each configuration.

Table 2: Comparison of accuracies and generalization metrics across datasets and model architectures. Bold: best, Italic: second-best.

| Dataset | Metric | None | ElasticNet | $\psi_{spec}$ |
|---|---|---|---|---|
| PhilUSIL | Train Acc (%) | 96,60 | 96,04 | 97,28 |
| | Val Acc (%) | 94,92 | 94,98 | 95,81 |
| | Test Acc (%) | 93,73 | 94,36 | 94,27 |
| | Δ Train-Val (%) | 1,68 | 1,06 | 1,47 |
| | Δ Val-Test (%) | 1,19 | 0,62 | 1,54 |
| | Δ Train-Test (%) | 2,87 | 1,68 | 3,01 |
| | GES | 0.0000 | 0.1605 | -0.1164 |
| | GenScore | 0.0000 | 1.0000 | 0.0213 |
| Waveform | Train Acc (%) | 88,80 | 87,43 | 89,74 |
| | Val Acc (%) | 86,80 | 85,07 | 85,07 |
| | Test Acc (%) | 86,27 | 84,67 | 84,67 |
| | Δ Train-Val (%) | 2,00 | 2,36 | 4,67 |
| | Δ Val-Test (%) | 0,53 | 0,40 | 0,40 |
| | Δ Train-Test (%) | 2,53 | 2,76 | 5,07 |
| | GES | 0.0000 | -0.0143 | -0.1577 |
| | GenScore | 0.0000 | 1.0000 | 0.1590 |
| BBC News | Train Acc (%) | 95,17 | 95,03 | 95,91 |
| | Val Acc (%) | 94,84 | 94,57 | 94,84 |
| | Test Acc (%) | 93,21 | 93,21 | 93,48 |
| | Δ Train-Val (%) | 0,33 | 0,46 | 1,07 |
| | Δ Val-Test (%) | 1,63 | 1,36 | 1,36 |
| | Δ Train-Test (%) | 1,96 | 1,82 | 2,43 |
| | GES | 0.0000 | 0.1321 | -0.4448 |
| | GenScore | 0.0000 | 1.0000 | 0.1689 |
| MNIST | Train Acc (%) | 97,87 | 96,72 | 97,45 |
| | Val Acc (%) | 95,36 | 96,44 | 97,38 |
| | Test Acc (%) | 95,22 | 96,03 | 95,58 |
| | Δ Train-Val (%) | 2,51 | 0,28 | 0,07 |
| | Δ Val-Test (%) | 0,14 | 0,41 | 1,80 |
| | Δ Train-Test (%) | 2,65 | 0,69 | 1,87 |
| | GES | 0.0000 | 0.1254 | 0.9573 |
| | GenScore | 0.0000 | 1.0000 | 0.3023 |
| HAR | Train Acc (%) | 92,87 | 90,89 | 92,68 |
| | Val Acc (%) | 90,79 | 90,50 | 91,11 |
| | Test Acc (%) | 89,00 | 90,33 | 89,32 |
| | Δ Train-Val (%) | 2,08 | 0,39 | 1,57 |
| | Δ Val-Test (%) | 1,79 | 0,17 | 1,79 |
| | Δ Train-Test (%) | 3,87 | 0,56 | 3,36 |
| | GES | 0.0000 | 0.0251 | 0.4238 |
| | GenScore | 0.0000 | 1.0000 | 0.2653 |
| 20 Newsgroups | Train Acc (%) | 94,91 | 93,99 | 94,51 |
| | Val Acc (%) | 88,98 | 90,55 | 93,75 |
| | Test Acc (%) | 88,47 | 88,65 | 88,74 |
| | Δ Train-Val (%) | 5,93 | 2,84 | 0,76 |
| | Δ Val-Test (%) | 0,51 | 1,90 | 5,01 |
| | Δ Train-Test (%) | 6,44 | 4,74 | 5,77 |
| | GES | 0.0000 | 0.9549 | 2.6193 |
| | GenScore | 0.0000 | 1.0000 | 0.3956 |

| Dataset | Metric | None | Elastic Net | $\psi_{spec}$ |
|---|---|---|---|---|
| GSC v2 | Train Acc (%) | 88,23 | 87,42 | 87,20 |
| | Val Acc (%) | 87,59 | 86,99 | 86,79 |
| | Test Acc (%) | 84,96 | 86,33 | 84,40 |
| | Δ Train-Val (%) | 0,64 | 0,43 | 0,41 |
| | Δ Val-Test (%) | 2,63 | 0,66 | 2,39 |
| | Δ Train-Test (%) | 3,27 | 1,09 | 2,80 |
| | GES | 0.0000 | 0.2951 | 0.8156 |
| | GenScore | 0.0000 | 1.0000 | 0.6781 |
| GTSRB | Train Acc (%) | 92,20 | 87,83 | 91,78 |
| | Val Acc (%) | 92,07 | 86,48 | 89,21 |
| | Test Acc (%) | 87,51 | 85,68 | 87,40 |
| | Δ Train-Val (%) | 0,13 | 1,35 | 2,57 |
| | Δ Val-Test (%) | 4,56 | 0,80 | 1,81 |
| | Δ Train-Test (%) | 4,69 | 2,15 | 4,38 |
| | GES | 0.0000 | 0.3394 | 0.2163 |
| | GenScore | 0.0000 | 1.0000 | 0.1265 |
| ESC-50 | Train Acc (%) | 78,07 | 78,21 | 76,36 |
| | Val Acc (%) | 78,00 | 75,33 | 76,00 |
| | Test Acc (%) | 73,67 | 74,33 | 74,00 |
| | Δ Train-Val (%) | 0,07 | 2,88 | 0,36 |
| | Δ Val-Test (%) | 4,33 | 1,00 | 2,00 |
| | Δ Train-Test (%) | 4,40 | 3,88 | 2,36 |
| | GES | 0.0000 | 0.1192 | 1.8629 |
| | GenScore | 0.0000 | 0.3729 | 1.0000 |
| CIFAR100 | Train Acc (%) | 75,47 | 78,04 | 73,40 |
| | Val Acc (%) | 74,85 | 75,00 | 72,54 |
| | Test Acc (%) | 69,77 | 72,66 | 70,46 |
| | Δ Train-Val (%) | 0,62 | 3,04 | 0,86 |
| | Δ Val-Test (%) | 5,08 | 2,34 | 2,08 |
| | Δ Train-Test (%) | 5,70 | 5,38 | 2,94 |
| | GES | 0.0000 | 0.3201 | 2.1156 |
| | GenScore | 0.0000 | 0.2180 | 1.0000 |
| Stanford Cars | Train Acc (%) | 83,72 | 86,84 | 85,53 |
| | Val Acc (%) | 82,79 | 85,24 | 84,44 |
| | Test Acc (%) | 77,82 | 84,53 | 81,46 |
| | Δ Train-Val (%) | 0,93 | 1,60 | 1,09 |
| | Δ Val-Test (%) | 4,97 | 0,71 | 2,98 |
| | Δ Train-Test (%) | 5,90 | 2,31 | 4,07 |
| | GES | 0.0000 | 0.3332 | 2.8833 |
| | GenScore | 0.0000 | 0.0964 | 1.0000 |

| Dataset | Metric | None | Elastic Net | $\psi_{spec}$ |
|---|---|---|---|---|
| FoodX-251 | Train Acc (%) | 69,53 | 67,41 | 65,19 |
| | Val Acc (%) | 65,39 | 64,51 | 63,44 |
| | Test Acc (%) | 62,99 | 63,80 | 62,81 |
| | Δ Train-Val (%) | 4,14 | 2,90 | 1,75 |
| | Δ Val-Test (%) | 2,40 | 0,71 | 0,63 |
| | Δ Train-Test (%) | 6,54 | 3,61 | 2,38 |
| | GES | 0.0000 | 0.2287 | 0.2517 |
| | GenScore | 0.0000 | 0.7660 | 1.0000 |
| LWR | Train Acc (%) | 80,71 | 81,89 | 79,18 |
| | Val Acc (%) | 79,61 | 80,13 | 79,11 |
| | Test Acc (%) | 76,85 | 78,57 | 77,37 |
| | Δ Train-Val (%) | 1,10 | 1,76 | 0,07 |
| | Δ Val-Test (%) | 2,76 | 1,56 | 1,74 |
| | Δ Train-Test (%) | 3,86 | 3,32 | 1,81 |
| | GES | 0.0000 | 0.3481 | 1.6188 |
| | GenScore | 0.0000 | 0.5765 | 1.0000 |
| CAS-VSR-W1k | Train Acc (%) | 49,96 | 50,49 | 48,22 |
| | Val Acc (%) | 44,37 | 39,48 | 46,08 |
| | Test Acc (%) | 35,29 | 36,29 | 34,53 |
| | Δ Train-Val (%) | 5,59 | 11,01 | 2,14 |
| | Δ Val-Test (%) | 9,08 | 3,19 | 11,55 |
| | Δ Train-Test (%) | 14,67 | 14,20 | 13,69 |
| | GES | 0.0000 | 0.3353 | 8.7198 |
| | GenScore | 0.0000 | 0.4892 | 1.0000 |
| WebVision2 | Train Acc (%) | 62,91 | 59,79 | 61,96 |
| | Val Acc (%) | 61,06 | 56,98 | 61,43 |
| | Test Acc (%) | 51,26 | 56,51 | 57,65 |
| | Δ Train-Val (%) | 1,85 | 2,81 | 0,53 |
| | Δ Val-Test (%) | 9,80 | 0,47 | 3,78 |
| | Δ Train-Test (%) | 11,65 | 3,28 | 4,31 |
| | GES | 0.0000 | 0.1750 | 10.1245 |
| | GenScore | 0.0000 | 0.5882 | 1.0000 |
| iNaturalist2021 | Train Acc (%) | 69,24 | 70,43 | 68,68 |
| | Val Acc (%) | 64,15 | 60,14 | 64,31 |
| | Test Acc (%) | 57,09 | 59,37 | 59,24 |
| | Δ Train-Val (%) | 5,09 | 10,29 | 4,37 |
| | Δ Val-Test (%) | 7,06 | 0,77 | 5,07 |
| | Δ Train-Test (%) | 12,15 | 11,06 | 9,44 |
| | GES | 0.0000 | 0.0553 | 5.9493 |
| | GenScore | 0.0000 | 0.3107 | 1.0000 |

## 4.1 GENERALIZATION METRICS

### 4.1.1 Δ TRAIN-VAL, Δ VAL-TEST AND Δ TRAIN-TEST

To quantify the amount of generalization achieved by the learned function, we report three primary gap metrics in addition to standard Train, Validation, and Test accuracies. The Intermediate Generalization Gap (Δ Train-Val) measures the difference between training and validation accuracy, while the True Generalization Gap (Δ Train-Test) measures the difference between training and fully unseen test data. We also present the Validation-Test Gap (Δ Val-Test), which captures the difference between validation and test accuracy. Smaller values in all these gaps indicate better generalization, as the model maintains consistent performance out-of-sample and avoids overfitting. Conceptually, the Δ Train-Val provides an estimate of how well the model generalizes during training and hyperparameter tuning, the Δ Train-Test evaluates true generalization on completely unseen data, and the Δ Val-Test reflects the reliability of validation performance as a proxy for final test accuracy. A

large $\Delta$ Val-Test may suggest overfitting to the validation set or poor alignment between validation and test distributions.

### 4.1.2 GENERALIZATION EFFICIENCY SCORE

To more precisely quantify generalization quality beyond raw gap reductions (i.e., the simple difference between training and test accuracy), we introduce the Generalization Efficiency Score (GES). This metric jointly captures the extent to which a model reduces its generalization gap and retains validation and test accuracy, relative to an unregularized baseline. GES thus penalizes models that trivially reduce overfitting by underfitting, and favors those that achieve meaningful generalization improvements while maintaining high performance on unseen data. A detailed derivation and motivation of GES is provided in Appendix E.

### 4.1.3 GENSCORE

To better quantify generalization, we additionally report the Generalization Score (GenScore), a variance-normalized metric that assesses the smoothness of accuracy degradation from training through validation to test. Unlike simple accuracy ratios, GenScore adaptively weights performance gaps based on their empirical stability across models. Detailed construction and properties of Gen-Score are provided in Appendix F.

### 4.2 SUMMARY OF REGULARIZATION PERFORMANCE

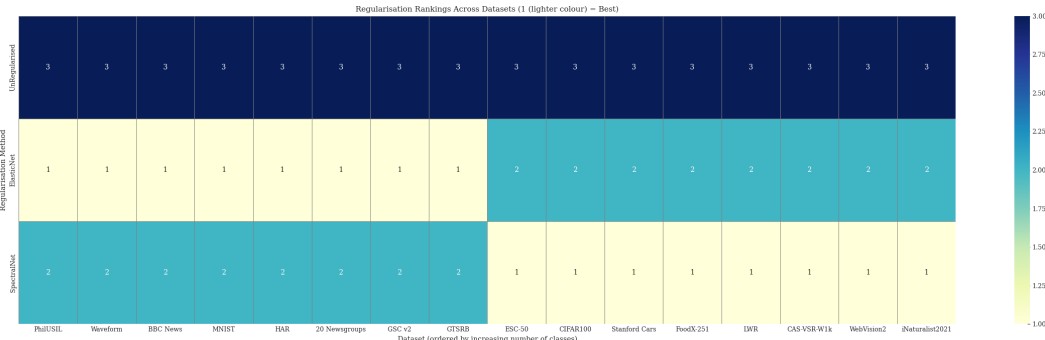

Figure 1: Heatmap of the ranking function $r : \mathcal{M} \times \mathcal{D} \to \{1, 2, 3\}$ for the regularization methods $\mathcal{M} = \{\text{UnRegularised}, \text{ElasticNet}, \text{SpectralNet}\}$ across the ordered datasets $\mathcal{D} = \{D_i\}_{i=1}^{16}$. Each entry $r(m, D_i)$ denotes the discrete rank of method $m$ on dataset $D_i$, where $r(m, D_i) = 1$ indicates the best-performing method and $= 3$ the worst. These rankings are derived by applying a hard threshold to the GenScore values in Table 2, converting them into discrete ranks to enable a clear comparative evaluation.

From Figure 1 , we can see that $\psi_{\text{spec}}$ consistently attains the best rank on large class size datasets, underscoring its usefulness in big class size settings. ElasticNet, by contrast, beats it on datasets with fewer classes. The Unregularised baseline uniformly ranks last, which is expected and validates the experimental setup, since setting the regularization parameter $\lambda = 0$ recovers this method in both cases. For a clearer view of these trends, we refer to Appendix G, where Figure 8 shows the trend in GenScore for $\psi_{\text{spec}}$ and ElasticNet as the number of classes grows and Figure 9 presents a more granular analysis of the effects of different regularisers $L^1$, $L^2$, and our proposed $\psi_1$ and $\psi_2$ by examining the $\alpha$ values. These results collectively demonstrate that $\psi_{\text{spec}}$ offers a new and useful regularization approach, particularly in large class size settings, beyond the regime where traditional methods remain competitive.

From Figure 2, the moving average of the GES, reveals distinct performance profiles for each regularization method. The unregularised baseline shows a moving average that is stable at 0. This outcome is not unexpected, as the Unregularised method serves as the baseline ($G_0$), and its GES is therefore fixed at 0 by construction. This provides a crucial and stable point of reference, validating the integrity of our experimental setup. The ElasticNet method maintains a moving average

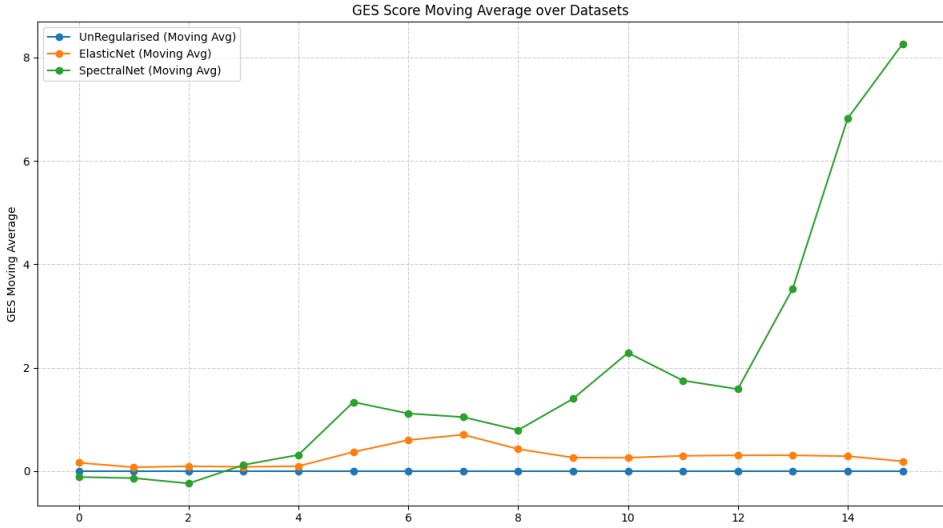

Figure 2: Comparison of the 3-point moving average of the Generalization Efficiency Score (GES) for Unregularised, ElasticNet, and $\psi_{\mathrm{spec}}$ methods. GES is a composite metric that rewards models for simultaneously reducing the generalization gap, retaining high accuracy, and maintaining stable validation-test performance as detailed in Appendix E. This smoothed visualization clearly reveals the distinct performance profiles of each regularization method as dataset class size (indexed from 1 to 16 as per Table 1 and as detailed in Figure 1's caption, i.e same Y axis) increases.

GES that remains consistently above baseline. This suggests that while it may satisfy some of the GES's constituent criteria, its overall contribution is limited by something, i.e it struggles to consistently achieve a substantial positive score implying that it either does not sufficiently reduce the generalization gap or fails to retain accuracy in a manner that yields a meaningful product across all three GES factors. In contrast, $\psi_{\mathrm{spec}}$ demonstrates a compelling and non-trivial performance. It has a poor early performance detailed by it's negative value but its moving average GES exhibits a pronounced upward trajectory, particularly as we transition to datasets of higher complexity. This behaviour is likely directly attributable to its regularization mechanism. The method's effectiveness becomes increasingly evident as the datasets grow in number of classes. For further details, refer to Figure 10, in Appendix G, which provides a point-by-point view of the GES scores on a logarithmic scale against class size, highlighting the marked fluctuations that characterize the performance of each method prior to smoothing. In these scenarios, the ability of $\psi_{\mathrm{spec}}$ to simultaneously reduce the generalization gap, retain high accuracy, and ensure stable performance on held-out data becomes a powerful asset, which the adaptive components of the GES metric keenly reward. This improvement provides strong empirical support for our hypothesis that characteristic function based regularization is a potent tool for achieving reliable generalization in challenging, large-scale settings.

## 5 CONCLUSION

In this work, we introduced a new class of regularization based on the characteristic function. We demonstrated a specific implementation of this framework by modeling the output layer of a neural network as a function of random variables and provided the necessary theoretical proofs to validate its properties. Our empirical experiments on a diverse selection of real-world datasets yielded results that directly align with our theoretical claims: the empirical data suggests that our method has a robust scaling behaviour, where the it's benefit becomes increasingly pronounced as the class size grows. These compelling results demonstrate that our characteristic function-based regularization is a promising method for future exploration and use, especially given the trend toward large-scale, high-parameter models, where its ability to promote another layer of robust generalization could prove to be greatly useful.

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

# A APPENDIX

## A.1 SATISFACTION OF LYAPUNOV CONDITION

**Definition 5.** Suppose there exists a sequence of independent random variables $\{Y_1, Y_2, ...Y_n\}$, with finite mean and variance, we can expect that the growth of the moments are limited by the Lyapunov Condition.

$$\lim_{n \to \infty} \frac{1}{s_n^{2+\delta}} \sum_{i=1}^{n} \mathbb{E}\left[|Y_i - \mu_i|^{2+\delta}\right] = 0 \tag{20}$$

**Definition 6.** For some sequence of independent Bernoulli random variables $\{X_1, X_2, ..X_n\}$, such that

$$X_i \sim Bernoulli(p_i) \tag{21}$$

$\mathbb{P}(X_i = 1) = p, 0 \le p \le 1, E(X_i) = p, Var(X_i) = p_i(1 - p_i)$

**Proposition 2.** Under most conditions, the Lyapunov CLT condition holds for Bernoulli Random Variables.

*Proof.*

$$\lim_{n \to \infty} \frac{1}{s_n^{2+\delta}} \sum_{i=1}^{n} E[(|X_i - \mu_i|^{2+\delta})] \tag{22}$$

Without Loss of Generality, let $\delta = 2$:

$$\lim_{n \to \infty} \frac{1}{s_n^4} \sum_{i=1}^{n} E[(|X_i - \mu_i|^4)] \tag{23}$$

By replacing $\mu_i$ with $E(X_i)$:

$$= \lim_{n \to \infty} \frac{1}{s_n^4} \sum_{i=1}^{n} E[(|X_i - E(X_i)|^4)] \tag{24}$$

By the Law of the Unconscious Statistician (LOTUS):

$$= \lim_{n \to \infty} \frac{1}{s_n^4} \sum_{i=1}^{n} \sum_{X_i=0}^{X_i=1} [(|X_i - E(X_i)|^4)] \tag{25}$$

By definition of Bernoulli distribution:

$$= \lim_{n \to \infty} \frac{1}{s_n^4} \sum_{i=1}^{n} (0 - p_i)^4 (1 - p_i) + (1 - p_i)^4 (p_i) \tag{26}$$

With reference to equation 2:

$$= \lim_{n \to \infty} \frac{1}{(\sum_{i=1}^{n} \sigma^2)^2} \sum_{i=1}^{n} p_i^4 (1 - p_i) + (1 - p_i)^4 (p_i) \tag{27}$$

By the Variance described for Bernoulli Random Variables, $\sigma^2 = p_i(1 - p_i)$:

$$= \lim_{n \to \infty} \frac{1}{(\sum_{i=1}^{n} (p_i(1 - p_i)))^2} \sum_{i=1}^{n} p_i^4 (1 - p_i) + (1 - p_i)^4 (p_i) \tag{28}$$

Since parameter $0 \le p \le 1$, we can claim $p_i^4 \le p_i$ and $(1 - p_i)^4 \le (1 - p_i)$:

$$\le \lim_{n \to \infty} \frac{1}{(\sum_{i=1}^{n} (p_i(1 - p_i)))^2} \sum_{i=1}^{n} p_i(1 - p_i) + (1 - p_i)(p_i) \tag{29}$$

$$= \lim_{n \to \infty} \frac{1}{(\sum_{i=1}^{n} (p_i(1 - p_i)))^2} \sum_{i=1}^{n} 2p_i(1 - p_i) \tag{30}$$

By Linearity of the Sum,

$$= \lim_{n \to \infty} \frac{2\sum_{i=1}^{n} (p_i(1 - p_i))}{(\sum_{i=1}^{n} (p_i(1 - p_i)))^2} \tag{31}$$

$$= \lim_{n \to \infty} \frac{2}{\sum_{i=1}^{n} (p_i(1 - p_i))} \tag{32}$$

As $n \to \infty$,

$$\because \sum_{i=1}^{n} (p_i(1 - p_i)) \to \infty \tag{33}$$

We have

$$\lim_{n \to \infty} \frac{2}{\sum_{i=1}^{n} (p_i(1 - p_i))} = 0 \tag{34}$$

as desired. $\qquad\square$

## B  DISTANCE MEASURES

In this section, we extend the concept of $L_p$ norms to measure the differences between the distributions $\phi_D$ and $\phi_{\mathcal{N}(0,1)}$. We define the distance function $d(\phi_D, \phi_{\mathcal{N}(0,1)})$ by calculating the pointwise differences between the two distributions and applying the $L_p$ norms.

We start with the general definition of the $L_p$ norm for a vector $\mathbf{x} = (x_1, x_2, \ldots, x_n)$:

$$||\mathbf{x}||_p = \left( \sum |x_k|^p \right)^{\frac{1}{p}}, \quad p \ge 1. \tag{35}$$

Extending the definition of the standard $L_1$ norm, which provides a measure based on the absolute differences:

$$\psi_1 = d_1(\phi_D, \phi_{\mathcal{N}(0,1)}) = ||\phi_D - \phi_{\mathcal{N}(0,1)}||_1 = \sum_{k=-\infty}^{\infty} |\phi_D(u_k) - \phi_{\mathcal{N}(0,1)}(u_k)|. \tag{36}$$

Next, we extend the $L_2$ norm, which measures the Euclidean distance between the pointwise differences:

$$\psi_2 = d_2(\phi_D, \phi_{\mathcal{N}(0,1)}) = ||\phi_D - \phi_{\mathcal{N}(0,1)}||_2 = \sqrt{\sum_{k=-\infty}^{\infty} |\phi_D(u_k) - \phi_{\mathcal{N}(0,1)}(u_k)|^2}. \tag{37}$$

Finally, we define a convex combination of $\psi_1$ and $\psi_2$, analogous to the ElasticNet regularization. This formulation, we call SpectralNet[2], allows interpolation between the $\psi_1$ and $\psi_2$ :

$$\psi_{\text{spec}} = d_{\text{spec}}(\phi_D, \phi_{\mathcal{N}(0,1)}) = \alpha\,\psi_1 + (1-\alpha)\,\psi_2, \quad \alpha \in [0,1]. \tag{38}$$

Here, $\alpha$ governs the trade-off: $\alpha = 1$ recovers the pure $\psi_1$ distance, while $\alpha = 0$ corresponds to the $\psi_2$ distance. Intermediate values yield a hybrid measure, capturing both distributional discrepancies.

While geometric distance measures could potentially yield greater performance, we have chosen to focus on these straightforward metrics to provide a gentle introduction to the topic and methodology discussed in this paper.

## C    CHARACTERISTIC FUNCTION SIMULATION FIGURES

### C.1    CHARACTERISTIC FUNCTION OF NORMAL AND BERNOULLI DISTRIBUTION

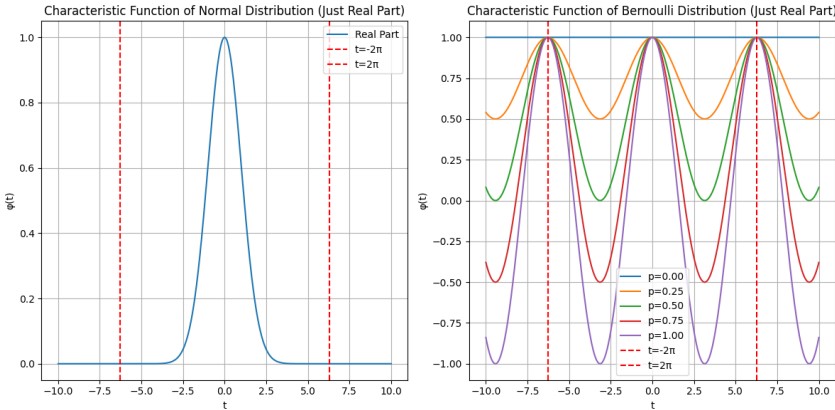

Figure 3: Plot of Normal and Bernoulli Characteristic Functions (Only Real Part)

The figure 3 shows the plot of real part of the Normal and Bernoulli Distribution. We thought this would be apt to add as this is to give a visual intuition for the reader for how these functions look when graphed as there is not much literature regarding visualising them.

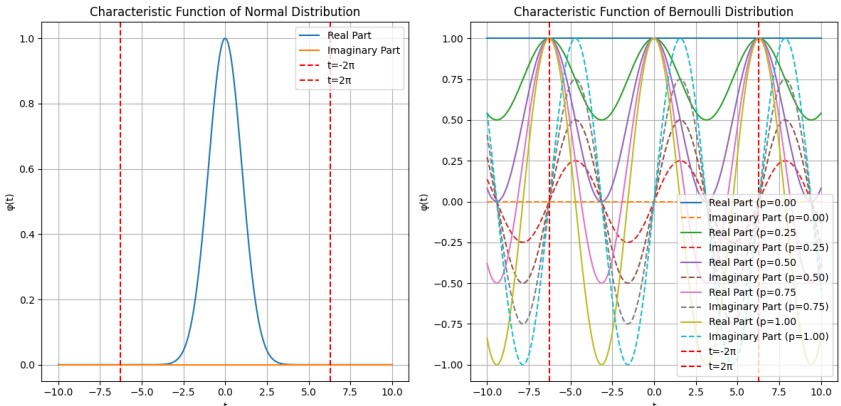

Figure 4: Extended Plot of Normal and Bernoulli Characteristic Function (Includes Imaginary Part)

### C.2 IMAGINARY PART INCLUSIVE CHARACTERISTIC FUNCTION OF NORMAL AND BERNOULLI DISTRIBUTION

The figure 4 shows the plot of the Normal and Bernoulli Distribution inclusive of the imaginary part. It is interesting to note the imaginary part is on the zero line for the Normal. As for the Bernoulli we can see a "phase" difference between the Imaginary and the Real Part.

If one would like to explore why, they can derive insight using the following as a starting point:

**Definition 7.** Euler's formula (Euler, 1748) (Cotes, 1714) states that for any real number $x$:

$$e^{\vartheta x} = \cos(x) + \vartheta \sin(x) \tag{39}$$

This formula can be used to express complex exponentials in terms of trigonometric functions.

**Definition 8.** Using equation 5 and definition 7, the characteristic function of a random variable $X$ is defined as:

$$\phi(u) = \mathbb{E}[e^{\vartheta u X}] = \mathbb{E}[\cos(uX)] + \vartheta \mathbb{E}[\sin(uX)] \tag{40}$$

where the real and imaginary parts of the characteristic function are:

$$\mathrm{Re}(\phi(tu) = \mathbb{E}[\cos(uX)] \tag{41}$$

$$\mathrm{Im}(\phi(u)) = \mathbb{E}[\sin(uX)] \tag{42}$$

### C.3 ZOOMED OUT VIEW TO OBSERVE PERIODICITY

The figure 5 shows that the Normal Characteristic Function does not seem to periodic unlike the Bernoulli Characteristic Function which seems to have a defined $\pi$-periodic structure It also shows that the Normal Characteristic Function is concentrated within the $-2\pi$ to $2\pi$ region. (Which motivated our choice in the numerics section D).

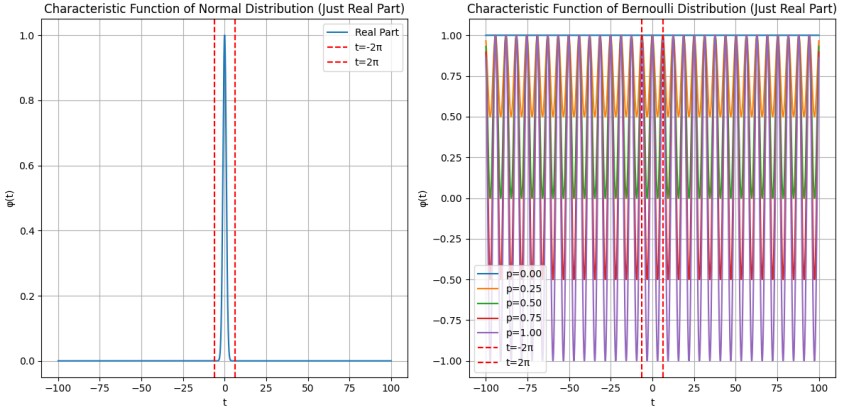

Figure 5: Plot of Normal and Bernoulli

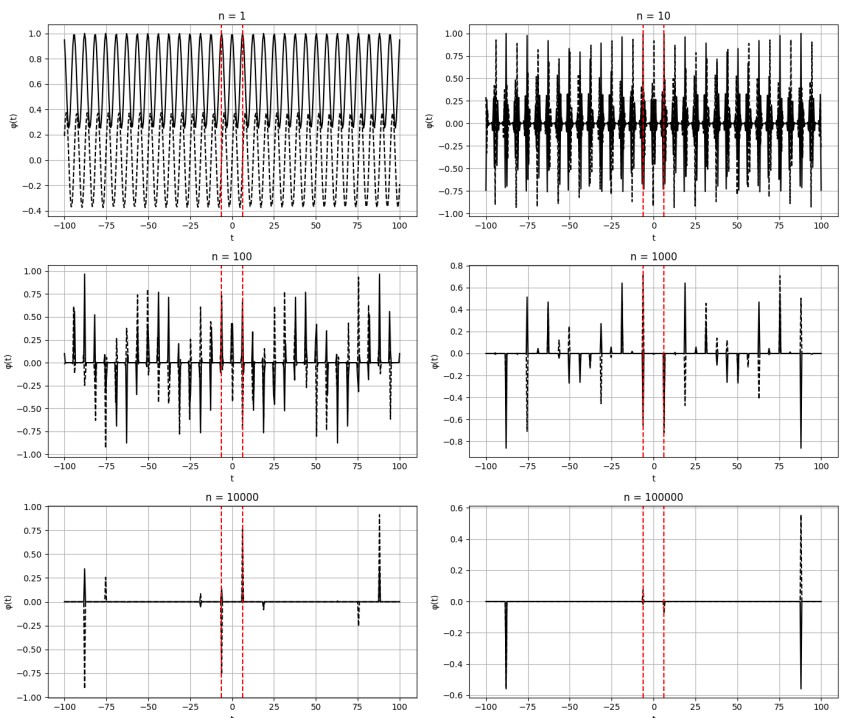

Figure 6: Numerical Simulation Plot of the Convergence

### C.4 BEHAVIOUR OF THE CHARACTERISTIC FUNCTION WHEN JUST ADDING BERNOULLI VARIABLES TOGETHER MINDLESSLY

The figure 6 is generated random generated $p_i$ values for a $\sum_{i=1}^{N}$ Bernoulli Distributions. It is interesting to note how just adding the Bernouli's will result in it resulting in a convergence towards the zero line.

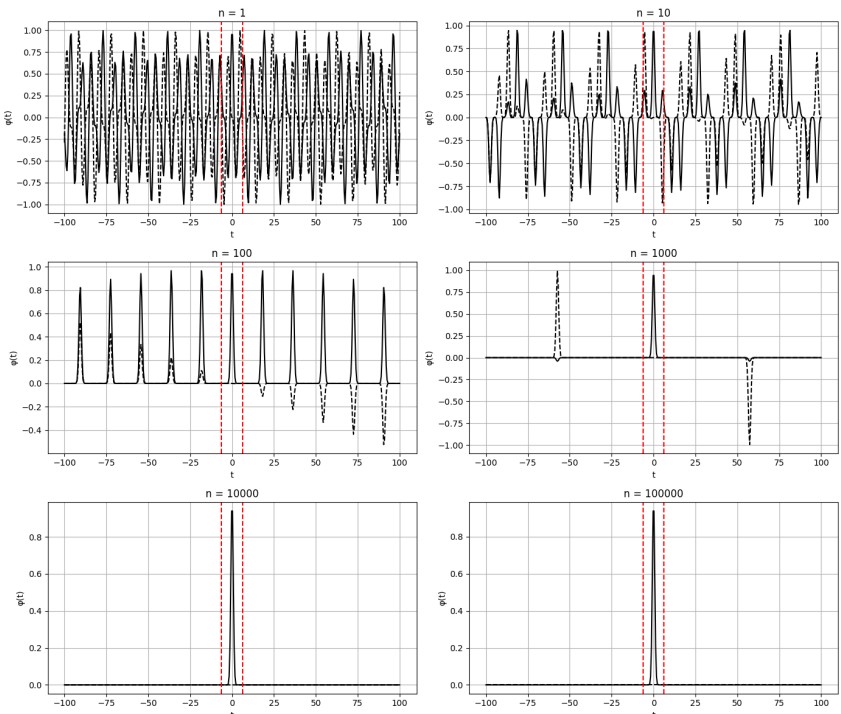

Figure 7: Numerical Simulation Plot of just adding Bernoullis

### C.5 NUMERICAL SIMULATION OF CONVERGENCE DESCRIBED IN PROPOSITION 1

The figure 7 is generated random generated $p_i$ values for a linear combination $N$ Bernoulli Distributions which are added according to the $\aleph$ model described in definition 3.

## D NUMERICAL CONSIDERATIONS

The characteristic function is generally a considered a "pure mathematical tool" whereby it's continuous nature presents significant challenges when implemented in discrete computational environments. Modern computers rely on finite precision arithmetic, which inherently restricts the exact representation of continuous functions, including characteristic functions. This means that we have to formulate discretizations based on some assumptions to integrate the characteristic function into a practical regularization algorithm for machine learning, enabling it to operate and execute within finite time.

Specifically, we will have to restrict it's domain to a "good enough" range since $t \in \mathbb{R}$ and the associated infinite nature of the reals. In other words, abstractly the problem is then as follows (with the help of some informal proof sketches for the sake of brevity due to page limit and for the reader's sanity) :

**Proposition 3.** The set of real numbers $\mathbb{R}$ is uncountably infinite.

**Informal Proof Sketch 1.** This can be shown using Cantor's famous Diagonal Argument. Assume for contradiction that $\mathbb{R}$ is countable. Then we can list all real numbers in the interval $[0, 1]$ as $r_1, r_2, r_3, \ldots$. We can then construct a new real number $r$ by taking the diagonal of this list and changing each digit, ensuring that $r$ differs from each $r_n$ at the $n$-th digit. Therefore, $r$ cannot be in our original list, contradicting the assumption that we had listed all real numbers. Thus, $\mathbb{R}$ is uncountably infinite. (Cantor, 1932)

---

[2]We refer to this as "Spectral" since the distance is constructed from characteristic functions, which are Fourier transforms of probability distributions. This naturally evokes the idea of "spectral concepts" into play especially in light of the term "eigen" (German for "characteristic") and hence motivates the name.

**Proposition 4.** The set of real numbers $\mathbb{R}$ is complete.

**Informal Proof Sketch 2.** The completeness of $\mathbb{R}$ can be demonstrated using Dedekind's cuts. A Dedekind cut partitions the rational numbers into two non-empty sets $A$ and $B$, where all elements of $A$ are less than all elements of $B$. For any non-empty set of rationals that is bounded above, there exists a least upper bound (supremum) in $\mathbb{R}$. This property ensures that every Cauchy sequence of real numbers converges to a real number, establishing the completeness of $\mathbb{R}$. (Dedekind, 2012)

**Proposition 5.** The set of computable real numbers is countably infinite.

**Informal Proof Sketch 3.** The set of computable real numbers can be described as those numbers for which there exists a finite algorithm (Turing machine) that can produce their digits. Since the set of all finite algorithms is countable, it follows that the set of computable real numbers is also countable.(Bournez, 2024)(Weihrauch, 2012)

**Proposition 6.** The set of computable real numbers is not complete.

**Informal Proof Sketch 4.** To see this, consider the sequence of computable numbers defined by $r_n = \frac{1}{n}$, which converges to 0. Although 0 is a limit point of the sequence, it is not computable because there is no finite algorithm that can output the exact value of 0. This demonstrates that there exist Cauchy sequences of computable real numbers that do not converge to a computable limit, thereby showing that the set of computable real numbers is not complete. (Bournez, 2024)(Weihrauch, 2012)

It becomes evident that propositions 3, 4, 5, and 6 present significant challenges in computing the desired function $\phi(t)$ especially on a Discrete Dynamical System like the modern computer we use. To address this, we have adopted a strategy of restricting to a finite domain of $t \in [-2\pi, 2\pi]$ where we discretize this interval into $n = 1000$ finite segments, which can be easily accomplished using a linear space function such as `numpy.linspace` in python or similar methods on any modern programming languages.

The rationale for selecting the interval $[-2\pi, 2\pi]$ is motivated by the analysis of the figures 3 and 5 in the appendix of the characteristic function for the standard normal distribution, as well as the set of convergence graphs for the $\aleph$-modelled linear combinations of Bernoulli random variables observed in 7. The region of primary interest lies within this interval, and while any variations outside this interval may be potentially significant under certain circumstances, we can effectively treat them as an acceptable level of statistical noise, we are willing to quantified by some $\epsilon$. This allows us to disregard this noise in the context of testing viability, though it may come at the expense of some regularization "performance".

There is no universally "correct" range or sample size ($n$); however, for the purposes of our experimentation, we consider this choice to be sufficient.

## E   CRAFTING A COMPOSITE SCALAR MEASURE TO BETTER REFLECT GENERALIZATION GAP

### E.1   MOTIVATION AND OVERVIEW

We introduce the **Generalization Efficiency Score (GES)**, a scalar metric designed to evaluate how effectively a model generalizes beyond the training distribution. Traditional metrics such as the raw generalization gap, defined as the difference between training and test accuracy, can be misleading. For instance, underfitting models often show small generalization gaps despite poor test performance. Likewise, some overfit models may achieve high accuracy on held-out data despite exhibiting larger gaps.

GES addresses this by incorporating three key factors:

- Relative reduction in generalization gap compared to a baseline,
- Retention of accuracy on both validation and test sets, and
- A penalty for test-validation disagreement.

This composite metric provides a more holistic view of generalization by rewarding models that achieve smaller generalization gaps *and* maintain stable, high performance on unseen data.

## E.2 FORMAL DEFINITION

Let $\mathcal{M}_i$ denote a trained model indexed by $i \in \{0, 1, \ldots, N\}$, evaluated on the following datasets:

- $\mathcal{D}_{\text{train}}$: training set,
- $\mathcal{D}_{\text{val}}$: validation set,
- $\mathcal{D}_{\text{test}}$: test set.

We hence get accuracies[3] of model $\mathcal{M}_i$ on these sets:

$$\text{Acc}_{\text{train}}^{(i)}, \quad \text{Acc}_{\text{val}}^{(i)}, \quad \text{Acc}_{\text{test}}^{(i)}. \in [0, 100]$$

We define the generalization gap of model $\mathcal{M}_i$ as:

$$G_i := \text{Acc}_{\text{train}}^{(i)} - \text{Acc}_{\text{test}}^{(i)}.$$

Let $\mathcal{M}_0$ be a fixed *baseline model* (typically unregularized), with:

$$G_0 := \text{Acc}_{\text{train}}^{(0)} - \text{Acc}_{\text{test}}^{(0)}.$$

We now define the three components of GES:

**Gap Factor:** Measures improvement in generalization gap:

$$\text{GapFactor}_i := \frac{G_0 - G_i}{G_0}.$$

**Accuracy Retention Factor:** Measures retained accuracy relative to the baseline:

$$\text{AccFactor}_i := \alpha \cdot \frac{\text{Acc}_{\text{val}}^{(i)}}{\text{Acc}_{\text{val}}^{(0)}} + (1 - \alpha) \cdot \frac{\text{Acc}_{\text{test}}^{(i)}}{\text{Acc}_{\text{test}}^{(0)}}.$$

Here, $\alpha \in [0, 1]$ is a weight that balances validation and test accuracy. We compute it automatically based on the variance of test-validation discrepancies across models:

$$\alpha := 0.5 - 0.5 \cdot \frac{\text{Var}(d_1, \ldots, d_N)}{\text{Var}(d_1, \ldots, d_N) + \epsilon} \text{with } d_i := \left| \text{Acc}_{\text{test}}^{(i)} - \text{Acc}_{\text{val}}^{(i)} \right|,$$

where $R_{\text{val}}$ and $R_{\text{test}}$ are vectors of accuracies across models and $\epsilon = 10^{-6}$ is a small constant for numerical stability.

Intuitively, when the difference between test and validation accuracies exhibits low variance, the model performances are consistent, and we assign roughly equal weight to both accuracies ($\alpha \approx 0.5$). Conversely, if the discrepancy varies widely, indicating that validation accuracy may be unstable or less reliable, $\alpha$ shifts closer to 0, placing greater emphasis on the test accuracy, which is considered a more robust indicator of generalization. This adaptive weighting mechanism ensures the retention factor prioritizes the most trustworthy signal, yielding a more reliable measure of accuracy retention.

**Penalty Factor:** Penalizes disagreement between validation and test performance:

$$\text{Penalty}_i := \left( \left| \text{Acc}_{\text{test}}^{(i)} - \text{Acc}_{\text{val}}^{(i)} \right| \right)^k,$$

where the exponent $k$ is determined from the variance in discrepancies across all models:

$$k := 1 + \frac{\text{Var}(d_1, \ldots, d_N)}{\text{Var}(d_1, \ldots, d_N) + \epsilon}, \quad \text{with } d_i := \left| \text{Acc}_{\text{test}}^{(i)} - \text{Acc}_{\text{val}}^{(i)} \right|.$$

---

[3]All accuracy values are originally in the range $[0, 1]$, but are rescaled to the $[0, 100]$ range for metric computation. This scaling mitigates unintended numerical effects arising from exponentiation of values less than one, which can otherwise lead to disproportionately small quantities and instability.

The penalty factor quantifies the disagreement between validation and test accuracies for each model, raising this discrepancy to the power $k$ to modulate its influence. The exponent $k$ is adaptively determined based on the variance of these discrepancies across all models. When the variance is low, indicating consistent agreement between validation and test accuracies, $k$ remains close to 1, applying a moderate penalty. However, when the variance is high, reflecting unstable or divergent performance between validation and test sets, $k$ increases towards 2, amplifying the penalty on large discrepancies. This adaptive exponent ensures that models exhibiting greater inconsistency are penalized more heavily, thereby promoting reliability and stability in the overall scoring.

### E.3 FINAL METRIC

Combining the components, the Generalization Efficiency Score for model $\mathcal{M}_i$ is given by:

$$
\underbrace{\left( \frac{G_0 - G_i}{G_0} \right)}_{\text{Gap Improvement Factor}} \cdot \underbrace{\left( \alpha \cdot \frac{\text{Acc}_{\text{val}}^{(i)}}{\text{Acc}_{\text{val}}^{(0)}} + (1 - \alpha) \cdot \frac{\text{Acc}_{\text{test}}^{(i)}}{\text{Acc}_{\text{test}}^{(0)}} \right)}_{\text{Accuracy Retention Factor}} \cdot \underbrace{\left( \left| \text{Acc}_{\text{test}}^{(i)} - \text{Acc}_{\text{val}}^{(i)} \right| \right)^k}_{\text{Validation-Test Discrepancy Penalty}}
\tag{43}
$$

### E.4 INTERPRETATION AND PROPERTIES

This construction ensures that:

- $\text{GES}_0 = 0$ by design (the baseline model),
- $\text{GES}_i > 0$ indicates an improvement in generalization *and* accuracy,
- $\text{GES}_i < 0$ indicates worse generalization, worse accuracy, or unstable validation-test behavior,

The Gap Imporvement Factor quantifies improvement in overfitting, the Accuracy Retention Factor encourages retention of predictive power, and the Penalty discourages large discrepancies between validation and test accuracy, which could signal instability or poor generalization.

### E.5 EXAMPLE CALCULATION

Consider a baseline model $\mathcal{M}_0$ with:

$$
\text{Acc}_{\text{train}}^{(0)} = 0.9185, \quad \text{Acc}_{\text{val}}^{(0)} = 0.9061, \quad \text{Acc}_{\text{test}}^{(0)} = 0.8893.
$$

This gives $G_0 = 0.9185 - 0.8893 = 0.0292$.

Now consider a second model $\mathcal{M}_2$ with:

$$
\text{Acc}_{\text{train}}^{(2)} = 0.9300, \quad \text{Acc}_{\text{val}}^{(2)} = 0.9007, \quad \text{Acc}_{\text{test}}^{(2)} = 0.8953,
$$
$$
G_2 = 0.9300 - 0.8953 = 0.0347, \quad d_2 = 100 \cdot |0.8953 - 0.9007| = 0.54.
$$

Assuming $\alpha = 0.6$ and $k = 2$ (for illustration), we compute:

$$
\text{GapFactor}_2 = \frac{0.0292 - 0.0347}{0.0292} \approx -0.1884
$$
$$
\text{AccFactor}_2 = 0.6 \cdot \frac{0.9007}{0.9061} + 0.4 \cdot \frac{0.8953}{0.8893} \approx 0.5956 + 0.4028 = 0.9984
$$
$$
\text{Penalty}_2 = (0.54)^2 = 0.2916
$$

$$
\text{GES}_2 = (-0.1884) \cdot 0.9984 \div 0.2916 \approx -0.645
$$

A negative score reflects worse generalization gap and marginal validation/test mismatch compared to the baseline.

### E.6 APPLICATION IN OUR STUDY

We use the Generalization Efficiency Score as a diagnostic in ablation and regularization studies. By choosing the unregularized model as a reference, we can meaningfully assess which techniques truly enhance generalization, rather than simply reducing capacity or trivially lowering the generalization gap. GES helps us focus on models that maintain accuracy while improving robustness on unseen data.

## F  VARIANCE-NORMALIZED METRIC FOR GENERALIZATION PERFORMANCE

In assessing model generalization, prevailing metrics often reduce the intricate geometry of performance degradation across data splits to a scalar ratio, typically, test accuracy divided by training accuracy. While intuitive, such metrics fail to account for the asymmetric roles played by different evaluation regimes (training, validation, test), and more critically, for the *relative stabilities* of these regimes across candidate models or hyperparameter configurations.

We propose a variance-normalized generalization score, denoted as **GenScore**, that measures the smoothness and consistency of performance degradation from training to validation to test. Unlike traditional metrics, GenScore respects the empirical variance structure of model behavior across evaluation splits and adapts its penalization accordingly.

Let each model instance $m$ be associated with a triplet of accuracies as follows:

$$\mathbf{a}^{(m)} = (\mathrm{Acc}_{\mathrm{train}}, \mathrm{Acc}_{\mathrm{val}}, \mathrm{Acc}_{\mathrm{test}}),$$

with natural deltas defined as:

$$\Delta_{\mathrm{tv}}^{(m)} = \mathrm{Acc}_{\mathrm{train}} - \mathrm{Acc}_{\mathrm{val}}, \quad \Delta_{\mathrm{vt}}^{(m)} = \mathrm{Acc}_{\mathrm{val}} - \mathrm{Acc}_{\mathrm{test}}, \quad \Delta_{\mathrm{tt}}^{(m)} = \mathrm{Acc}_{\mathrm{train}} - \mathrm{Acc}_{\mathrm{test}}.$$

These quantities encode distinct aspects of generalization: $\Delta_{\mathrm{tv}}$ captures overfitting to training data, $\Delta_{\mathrm{vt}}$ reflects sensitivity to unseen but proximate data, and $\Delta_{\mathrm{tt}}$ represents end-to-end generalization collapse.

To define a principled aggregation of these deltas into a scalar score, we take a data-adaptive approach. Let $\sigma_{\mathrm{tv}}^2, \sigma_{\mathrm{vt}}^2, \sigma_{\mathrm{tt}}^2$ denote the sample variances of these deltas across a fixed set of model configurations $\mathcal{M}$, e.g., top-performing runs or a representative validation batch. We define inverse-variance weights:

$$w_i = \frac{\frac{1}{\sigma_i^2}}{\sum_j \frac{1}{\sigma_j^2}}, \quad i \in \{\mathrm{tv}, \mathrm{vt}, \mathrm{tt}\},$$

which serve as soft attention coefficients over the deltas, allocating greater influence to more stable differences. These weights reflect a natural statistical heuristic that stable gaps are more trustworthy indicators of systemic generalization behaviour.

The raw generalization penalty for model $m$ is then:

$$\mathcal{L}_{\mathrm{gen}}^{(m)} = w_{\mathrm{tv}} \cdot (\Delta_{\mathrm{tv}}^{(m)})^2 + w_{\mathrm{vt}} \cdot (\Delta_{\mathrm{vt}}^{(m)})^2 + w_{\mathrm{tt}} \cdot (\Delta_{\mathrm{tt}}^{(m)})^2,$$

which is minimized when performance degrades smoothly and uniformly. Finally, we normalize this penalty across $\mathcal{M}$ to obtain a bounded generalization score between [0,1] as follows:

$$\mathrm{GenScore}^{(m)} = 1 - \frac{\mathcal{L}_{\mathrm{gen}}^{(m)} - \min_{m' \in \mathcal{M}} \mathcal{L}_{\mathrm{gen}}^{(m')}}{\max_{m' \in \mathcal{M}} \mathcal{L}_{\mathrm{gen}}^{(m')} - \min_{m' \in \mathcal{M}} \mathcal{L}_{\mathrm{gen}}^{(m')} + \varepsilon},$$

with $\varepsilon = 10^{-6}$ ensuring numerical stability. This final score lies in the interval $[0, 1]$, with higher values indicating smoother and more robust generalization. Models with high training accuracy but erratic validation or test behavior are sharply penalized, while those that exhibit graceful degradation are rewarded.

We find that GenScore correlates more reliably with downstream robustness metrics (e.g., performance under distribution shift) than naïve ratios or accuracy gaps. Its variance-normalized structure allows it to adapt to the particular generalization geometry of the task, architecture, and dataset under study.

## G    FURTHER FIGURES

The figures in this section provide additional miscellaneous insight into the performance and internal behaviour of the different regularization strategies explored in the main text.

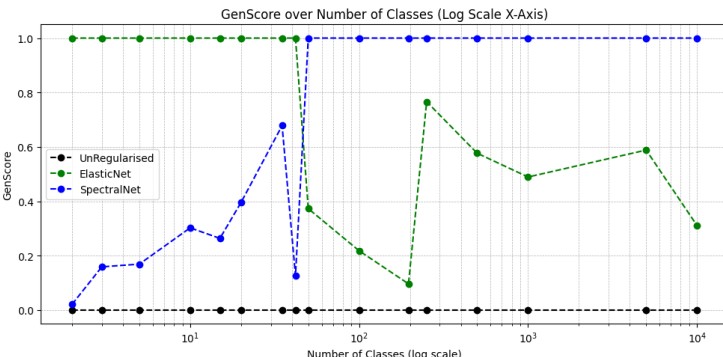

Figure 8: **GenScore over number of classes (log scale)**. This plot visualises the GenScore performance of three regularization methods (UnRegularised, ElasticNet, SpectralNet) across datasets ordered by increasing number of classes (x-axis in log scale). SpectralNet exhibits a clear upward trend, with GenScore increasing sharply and saturating at 1 as class cardinality increases. ElasticNet achieves the highest GenScore on low-class datasets, but its performance degrades on high-class settings. The Unregularised baseline remains consistently near zero across all datasets. This separation supports the conclusion that SpectralNet is uniquely effective in high-class regimes.

Figure 8 plots the GenScore values for Unregularised, ElasticNet, and SpectralNet methods across datasets ordered by increasing class cardinality (x-axis on a log scale). The results reveal a clear separation in behaviour, whereby ElasticNet dominates in low-class-count regimes, but its performance deteriorates as the number of classes increases. In contrast, SpectralNet exhibits a monotonic increase in GenScore, approaching a saturation point near 1 (implying it is the best performing after normalization). This suggests that SpectralNet is particularly well-suited for high-class scenarios. The Unregularised baseline remains at zero throughout, as expected. This plot underpins the discrete rankings shown in Figure 1 by offering the continuous underlying signal prior to thresholding and shows the second degree of the "variability" of the performance too where we can see "how much better or worse" each method is performing relative to each other and the baseline in the space of the GenScore as defined in Appendix F.

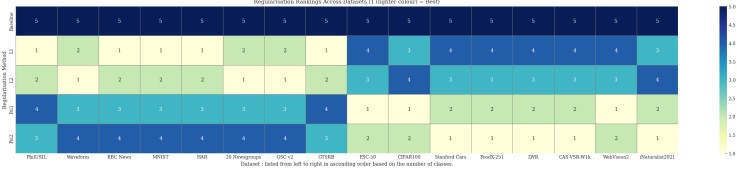

Figure 9: **Discrete ranking of regularization methods using learned $\alpha$ values.** This figure displays the ranking function $r(m, D_i)$ for five regularization strategies across datasets of increasing class complexity. The methods include Baseline ($\lambda = 0$), $L^1$, $L^2$, and the proposed $\psi_1$, $\psi_2$ regularisers. Rankings are derived from the relative magnitude of the learned regularization weight $\alpha$ assigned to each method within the same model. This granular perspective, derived from interpolation factor $\alpha$, offers an extra insight into the different regimes.

To complement the performance-based perspective, Figure 9 provides a more granular look at how different regularization terms influence the model internally. Here, we report the ranking of regularisers according to the linear interpolant coefficient $\alpha$ associated with each term, within a shared architecture. Specifically, we compare the standard $L^1$ and $L^2$ penalties with our proposed spectral regularisers $\psi_1$ and $\psi_2$. The results show that there is a skew towards $L^1$ and $L^2$ in simpler low class size tasks, while $\psi_1$ and $\psi_2$ emerge randomly dominant in datasets with many classes. This rank-

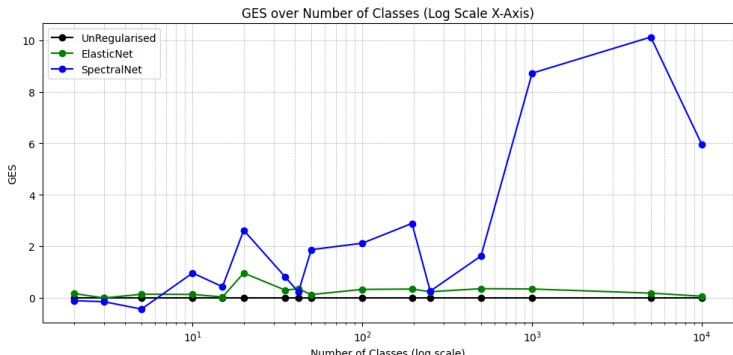

Figure 10: A point-by-point view of the Generalization Efficiency Score (GES) for Unregularised, ElasticNet, and SpectralNet methods. The GES is a composite metric that rewards models for simultaneously reducing the generalization gap, retaining high accuracy, and maintaining stable validation-test performance. This visualization highlights the distinct performance of each regularization method as dataset complexity, indexed by the number of classes, increases on a logarithmic scale.

ing reflects the model's own inductive bias, revealing which "flavour" of regularization it favours in different dataset settings. We caution, however, that these rankings should not be over-interpreted in isolation: in many cases, the top-ranked methods had $\alpha$ values close to others (e.g., near 0.5), indicating that the model viewed several regularisers as comparably useful. The ranking here reflects relative ordering rather than the magnitude of preference.

Figure 10 provides a greater analysis of the GES scores as a function of dataset complexity, which is indexed by the number of classes on a logarithmic scale. The Unregularised baseline, serving as our reference, demonstrates a GES that remains consistently at zero, obviously indicating a persistent lack of generalization across all class sizes as per definition of GES in Appendix E. The ElasticNet method exhibits a similar behavior, maintaining a GES that largely hovers above zero, with only minor fluctuations, suggesting a limited capacity to leverage its regularization for a more meaningful improvement in generalization as class sizes changes. In contrast, the performance of SpectralNet is markedly distinct. Its GES score is initially negative, indicating a worse performance than the baseline on datasets with a small number of classes, probably a consequence of the central limit theorem, i.e in this case we have not hit a sufficiently large enough sample in the "class size space". Thus, in ultra-low complexity settings, where simpler regularization is sufficient, our approach imposes an extremely poor form of regularization , leading to a negative GES[4]. However, its GES demonstrates a dramatic increase as the class count rises [5]. This trend underscores the unique efficacy of characteristic function-based regularization, whose ability to promote stability and reduce the generalization gap becomes increasingly pronounced and effective in high-dimensional settings where the sufficiently large class count satisfies the conditions for the Central Limit Theorem and allows the regularization to kick into gear more effectively.

---

[4]The Figure 7 in Appendix C can give a somewhat visual idea that explains in some way the discrepancy that happens in the ultra-low class size setting.

[5]Given the nature of the data, the observed increase in performance on datasets with a larger number of classes is likely due to the regularization's mechanism agreeing more favourably with the Central Limit Theorem assumption as the number of classes tends towards infinity.

