# OpenReview forum: "CharFxReg : Characteristic Function based Regularisation"
_ICLR.cc/2026/Conference — Submitted to ICLR 2026_

### Official Review · Reviewer_A6TH · 2025-10-19

**Soundness:** 2
**Presentation:** 1
**Contribution:** 1
**Rating:** 0
**Confidence:** 4

**Summary:**

The submission proposes a regularization method that involves matching the distribution of the (standardized) sum of instances of the output random variable of a predictive neural network model with that of a standard Normal, using the characteristic function representations. This is motivated by the Central Limit Theorem, which states that sums of a random variable converge to a Gaussian distribution.

Experiments show some improvements in generalization with respect to unregularized and ElasticNet models for a range of datasets and neural networks.

**Strengths:**

The key idea is intuitively sensible, and novel, to my knowledge.

The experiments are conducted over a wide range of datasets and models.

**Weaknesses:**

The primary weaknesses of the submission lies in adequately grounding the contributions in existing literature and in a robust experimental validation of the method.

  -- Regularization methods have been proposed in the past that also directly operate on the outputs of neural networks instead of weights, and the paper would be markedly strengthened if principled connections could be drawn to these earlier works, since it is likely that some of the existing methods have the same effect of "pulling in" overconfident predictive distributions: examples are label-smoothing, predictive-entropy regularization, spectral decoupling. Additionally, these are also more relevant to compare against, rather than solely ElasticNet.

 -- The experiments appear exhaustive at first sight, however the baseline numbers for almost all methods are quite lagging behind relative to where current state-of-the-art results are. On the one hand, one can argue that regularization effects are better demonstrated with room for improvement, but on the other, it seems pointless to showcase improvements on outdated baselines. Moreover, the performances are more often than not lagging behind the only point of comparison for regularizers, ElasticNet, which is hardly the most common regularizer unanimously used for such a wide range of models and datasets.

**Questions:**

No additional questions

---

### Official Review · Reviewer_Z2ha · 2025-10-26

**Soundness:** 2
**Presentation:** 1
**Contribution:** 1
**Rating:** 2
**Confidence:** 3

**Summary:**

This paper proposes a regularization technique based on the theory of characteristic function. The basic idea is to apply Lyapunov Central Limit Theorem on the model output (deep features in this paper). The characteristic function is derived to converge to the standard Gaussian distribution by applying the theorem. Then the paper proposes to explicitly drive the empirical output distribution estimated by training examples to the standard Gaussian, and uses their distance as a regularization term. Multiple experiments are conducted to evaluate the proposed method over diverse data modalities and models.  From the experimental results, the proposed method shows clear advantages in the Generalization Efficiency Score (GES) metric, but doesn’t seem to improve existing regularization techniques such as ElasticNet in terms of test accuracy.

**Strengths:**

1. Diverse datasets and models are used in the experiments.
2. The proposed method achieves best results in terms of GES.

**Weaknesses:**

1. The rationale of the regularization method is not quite clear. Since the characteristic function $\phi_{\mathcal{D}}$ already has the expected convergence to $\phi_{\mathcal{N}(0,1)}$ when n is large, why should we employ an additional regularization term constraining their distribution distance?
2. Implementation details are not described clearly, especially how the empirical term $\phi_{D}$ is calculated in model training. From Eq (7), it seems to also rely on $p_i$. Please provide source code or detailed pseudo code to clarify the steps and important hyperparameters.
3. The proposed method doesn’t show better experimental results in terms of test accuracy. The GES metric is reasonable, but test accuracy is still the most essential metric for most real applications.
4. Page 5 claims that ImageNet and ImageNet-21K are used for evaluation but results are not reported.
5. Figure 2 may not be the most appropriate way to present the corresponding result, since the X-axis (Dataset Index) does not represent a continuous or ordered sequence. Using a line chart implies temporal or sequential relationships between adjacent points, which is not the case here.

**Questions:**

1. What is the time and memory complexity of the proposed method?
2. In section 3.1 the authors explain the motivation of the work is from “the observation that, in many classification settings, the output of a neural network (especially under sigmoid or softmax activation) can be interpreted as a sequence of Bernoulli trials”. Is there any concrete evidence for this?

---

> ### Author Response · Authors · 2025-11-12
>
> Hi there, thanks for reading and the detailed comments!
>
> I think there are some misconceptions with regards to what is happening in the paper. I will give you a quick idea of what is going on followed by answering the stuff you mentioned to keep our discussion clearer.
>
> Idea of the paper:
> Essentially what is happening here in this paper is that it proposes the idea of regularizing based on characteristic functions whereby these always exist vs MGFs etc. It is NOT meant to be a drop in replacement for existing methods but instead it is another way to regularise. The idea here is that we present ONE "degenerate" way to do it by deriving a simple relationship between bernoullis and the normal distribution with regards to the class size and also presented the proof to which the empirical data backs up the given proof whereby what we expect is the following : As the class size increases (not the number of samples) -> we converge more and more to the given distribution , i.e we expect better GES performance and the empirical values do support this.
>
>
> regarding the things you mentioned:
> I will start by replying to the weakness followed by the questions , looking forward to hearing your thoughts on them after.
>
> 1) The rationale of the regularization method is not quite clear. Since the characteristic function $\phi_{\mathcal{D}}$ already has the expected convergence to $\phi_{\mathcal{N}(0,1)}$ when n is large, why should we employ an additional regularization term constraining their distribution distance?
>
> I think there is a misconception here regarding what is the n in question. From the way you phrase it , it seems like you think n refers to number of samples. In this case the n refers to the number of classes. So this regularization formulation presented is primarily focused on large class size problems and indeed it is shown theorectically to be true and does hold both empirically also as per the results.
>
> 2. Implementation details are not described clearly, especially how the empirical term $\phi_{D}$ is calculated in model training. From Eq (7), it seems to also rely on $p_i$. Please provide source code or detailed pseudo code to clarify the steps and important hyperparameters.
>
> I think the implementation should be rather obvious from construction and the appendix but I can see how it might be difficult to understand since I think there are some misconceptions present in your understanding of the paper. The p_i are just how you would interpret the output logits and then you are given the closed form expression of the value itself. Just take it as is and then use the appropriate distance function with respect to the phi_normal and then add this term to the loss with a lambda constant infront that is to be tuned depending on how much compute you want to throw at it.
>
> 3. The proposed method doesn’t show better experimental results in terms of test accuracy. The GES metric is reasonable, but test accuracy is still the most essential metric for most real applications.
>
> I would argue to say this is false when it comes to the idea of generalisation. As you mentioned, the GES metric is reasonable but I would say otherwise about the test accuracy as this was the point of the construction of GES to give a more detailed measure of the effect of regularisaton.  It is a more comprehensive metric/score that penalises deviation in generalisation behaviour compared to test accuracy which just gives you a limited idea of what is happening in your given the available samples. Experimentally what is being validated is whether this method works. It doesn't have to be SOTA in terms of test accuracy I think that is just chasing numbers and not the spirit of learning theory which is to actually look at things like these prove some bounds validate whether the expected behaviour holds true to some extent on an empricial setting which we observe in the paper.
>
> 4. Page 5 claims that ImageNet and ImageNet-21K are used for evaluation but results are not reported.
>
> Good catch . You are absolutely right, I realised some models when we were testing were pretrained on imagenet so I dropped these from evaluation but mistakenly left it there will rectify before the resubmit on 3rd Dec. Thank you for this kind point !
>
> 5. Figure 2 may not be the most appropriate way to present the corresponding result, since the X-axis (Dataset Index) does not represent a continuous or ordered sequence. Using a line chart implies temporal or sequential relationships between adjacent points, which is not the case here.
>
> I agree with you in spirit here but the idea instead here was to show the general idea of what I mentioned in point 1 and about how the method scales with increasing classes sizes where we expect this choice of method to do better as the number of classes increases. This graph is to show that improving trend is the expected result as we increase data set size during empirical experimentation. Hopefully that makes sense?
>
> Cont.

---

> ### Author Response · Authors · 2025-11-12
>
> contd.
>
> Regarding Questions :
>
> 1. What is the time and memory complexity of the proposed method?
> Should scale with output size, i.e number of classes.
>
> 2. I would say this is a really good question but it comes and I will answer from more philosophical angle cause I think it will be easier to understand. Essentially we can look at a output as Yes/no with respect to the class. Then when we do multiclass or single class classification we are just considering these sequences of yes nos to make our decision.
>
> Hope that clarifies! Looking forward to hearing what you think, really appreciate your perspective!

---

### Official Review · Reviewer_u9uM · 2025-10-28

**Soundness:** 1
**Presentation:** 1
**Contribution:** 2
**Rating:** 2
**Confidence:** 4

**Summary:**

This paper proposes "characteristic-function-based regularisation", which (1) models the outputs of a neural network by a chain of Bernoulli variables and (2) regularizes its characteristic function to be that of the unit normal distribution, based on the Lyapunov Central Limit Theorem of independent random variables. The proposed method is evaluated on a range of classification benchmarks that span the text, audio, and image domains.

**Strengths:**

Using characteristic functions to regularize neural networks is a novel idea to my knowledge.

**Weaknesses:**

The main weakness of the paper is that it does not really provide enough implementation details for the proposed method to be assessed. From my understanding, the paper treats the output activations of the neural networks as a weighted sum of independent Bernoulli variables. However, in Axiom 1 and Definition 3, it seems that $\mathscr{N}$ refers to "_data points_" sampled from the data distribution rather than network outputs, which is misleading. Moreover, it is not explained whether such treatment is applied to each output dimension (e.g., the classification probs of each class) independently or to the sum of all dimensions. It is neither clear to me how exactly the regularization term is calculated. Please see Questions for more details.

The motivation of the paper is also not very clear to me. Why could the output activations of neural networks be treated as a weighted combination of Bernoulli variables (except that they are indeed in the range $[0, 1]$)?

Finally, I do not think the current empirical results are significant enough to support the claim that the proposed method could improve generalization. Many of the gains on the proposed "generalization metrics" are simply due to the _decrease_ on train acc rather than the increase on test acc, which is meaningless in practice.

**Questions:**

- What does $\mathscr{N}$ exactly correspond to in a neural network?
- If $\mathscr{N}$ should asymptotically be unit Gaussian, why not directly regularize the distribution of $\mathscr{N}$ (e.g., by regularizing its moments) instead of resorting to its characteristic function?
- Why the output activations of a neural network should be modeled as a weighted combination of Bernoulli variables? What is the semantics of each Bernoulli variable?
- Appendix B: What are $u_k$ and $k$?

---

> ### Author Response · Authors · 2025-11-12
>
> Hi thanks for your comments.
> I will try to address it point by point and let me know your thoughts.
>
> Beginning with the questions :
> 1) N corresponds to the function/map equivalent to the output layer. (you mention it yourself in the summary (1) models the outputs of a neural network by a chain of Bernoulli variables) , a perfect network should indeed be such that you can generate a sample of the data points from the point of view of transforms from this.
>
> 2) Very very good point and yes this question is the core essence of why we developed this method! Yes you can choose to do so but the key idea of the paper and why we use characteristic functions is that there are some distributions which you don't have this direct relationship. A key point to note is that characteristic functions always exist, while MGFs might not. We chose the normal and bernouli as they are somewhat the degenerate case where you can easily check and understand what is going on and follow along. We feel using harder distributions and assumptions would be difficult for a reader to sort through as it might not be as easy to follow wrt the construction, given the ICLR page limit and also how much information is contained + the need for empirical validation it will indeed be unnecessarily terse and the idea won't translate as well. Also might be interesting to note MGF is quite related to the charcfx  . If you are familiar with the relationship with the fourier and laplace transform, it is just this in play :)
>
> 3) Again this is a by construction claim. You can choose to model it as you like but the underlying idea is that one way to interpret and build an output formulation that consists of neurons essentially saying "yes i am this" or "no I am not this" and then take a decision based on the combination of all these.  Hence why the choice as such.
>
> 4) These follow from definition of writing L1 norm of two measures as a sum.
>
> To address some other portions on your paragraph on the weakness :
>
>
> "Moreover, it is not explained whether such treatment is applied to each output dimension (e.g., the classification probs of each class) independently or to the sum of all dimensions. " . This should be clear from (3.3) line 16.
>
> I also want to clarify something regarding your claim that “a decrease in train accuracy rather than an increase in test accuracy is meaningless in practice.” I believe there is more to eval regularisation, and I’d like to explain why the behaviour you described is not only expected, but in fact desirable.
>
> Regularisation essentially acts as a smoothing mechanism on the model’s function where it intentionally reduces the model’s tendency to overfit by discouraging overly "complex" mappings that perfectly match the training data. When you introduce regularisation, you fundamentally alter the loss landscape. Because of this, it is almost never reasonable to expect the model to converge to the same training accuracy as before cause of a different objective.
>
> This change in the objective also means that, after training, variation in train and test accuracy can arise simply from the altered geometry of the loss surface. In such settings, the central goal is not to maximize either training or test accuracy directly, but rather to encourage tighter generalization gaps, that is, to make the model more stable and more “distribution-aware” so that it doesn’t overfit to noise or idiosyncrasies of the training set.
>
> Viewed this way, the behaviour captured by GES is exactly something we feel desirable to observe: they are designed to detect and quantify this smoothing effect, as an alteranative to reward raw test accuracy. Since, if we judged purely by test accuracy, we would miss the interesting regularization effect these methods aim to achieve.
>
> Ultimately, if you feel that elastic net can be accepted as a valid and effective regularizer, then our methods should be as well. We achieve similarish test performance, with the key distinction being that our approach exhibits precisely the desirable behaviour captured by the GES metric namely, reducing overfitting by smoothing the loss landscape and tightening the gap between train and test performance.
>
> The main idea here is to present a potential link between regularisation and using empirical characteristic functions as a potential way to work on larger class sizes.
>
> Please note, the empirical benchmarks provided are provided to serve as a way to show the methods general behaviour and less so as a SOTA type thing, it is purely in the spirit of learning theory and not a throw as much compute and tune hyperparameters till a perfect case is found. I feel theoretically they are rigorous enough and emprically they are sound enough to show the expected behaviour.
>
> Hence, the goal was to never "beat" elasticnet but rather present a new way to regularise.
>
> I appreciate the read and the qns you asked and hope these clarify most of your qns.
>
> Keen to hear what you think.

---

### Meta-Review · Area_Chair_4GZr · 2026-01-04

**Summary:**

The paper proposes a new regularisation procedure based on the characteristic function instead of L2 regularisation. The main issue with this paper is that the results are compared only with ElasticNet and without regularisation, and the results without regularisation are almost as good as those with the proposed method or with ElasticNet. Additionally, if we were to consider the horse race in terms of test accuracy, ElasticNet regularisation is typically superior (even when the differences are small). The paper require significant more work and justification before it can be accepted for publication at top ML conference.

**Reviewer Concerns:**

The quality of the results are poor and the reviewers were unimpressed by the theoretical justification.

**Reviewer Scores:**

The reviewers would not have modified their scores.

---

### Decision · Program_Chairs · 2026-01-26

Reject